# Collapsed variational Bayes for Markov jump processes

**Jiangwei Pan**[*][†]
Department of Computer Science
Duke University
panjiangwei@gmail.com

**Boqian Zhang**[*]
Department of Statistics
Purdue University
zhan1977@purdue.edu

**Vinayak Rao**
Department of Statistics
Purdue University
varao@purdue.edu

## Abstract

Markov jump processes are continuous-time stochastic processes widely used in statistical applications in the natural sciences, and more recently in machine learning. Inference for these models typically proceeds via Markov chain Monte Carlo, and can suffer from various computational challenges. In this work, we propose a novel collapsed variational inference algorithm to address this issue. Our work leverages ideas from discrete-time Markov chains, and exploits a connection between these two through an idea called uniformization. Our algorithm proceeds by marginalizing out the parameters of the Markov jump process, and then approximating the distribution over the trajectory with a factored distribution over segments of a piecewise-constant function. Unlike MCMC schemes that *marginalize* out transition times of a piecewise-constant process, our scheme *optimizes* the discretization of time, resulting in significant computational savings. We apply our ideas to synthetic data as well as a dataset of check-in recordings, where we demonstrate superior performance over state-of-the-art MCMC methods.

## 1  Markov jump processes

Markov jump processes (MJPs) (Çinlar, 1975) are stochastic processes that generalize discrete-time discrete-state Markov chains to continuous-time. MJPs find wide application in fields like biology, chemistry and ecology, where they are used to model phenomena like the evolution of population sizes (Opper and Sanguinetti, 2007), gene-regulation Boys et al. (2008), or the state of a computing network Xu and Shelton (2010). A realization of an MJP is a random piecewise-constant function of time, transitioning between a set of states, usually of finite cardinality $N$ (see Figure 1, left). This stochastic process is parametrized by an $N \times 1$ distribution $\pi$ giving the initial distribution over states, and an $N \times N$ rate matrix $A$ governing the dynamics of the process. The off-diagonal element $A_{ij}$ ($i \neq j$) gives the rate of transitioning from state $i$ to $j$, and these elements determine the diagonal element $A_{ii}$ according to the relation $A_{ii} = -\sum_{i \neq j} A_{ij}$. Thus, the rows of $A$ sum to 0, and the negative of the diagonal element $A_{ii}$ gives the *total* rate of leaving state $i$. Simulating a trajectory from an MJP over an interval $[0, T]$ follows what is called the Gillespie algorithm (Gillespie, 1977):

1. First, at time $t = 0$, sample an initial state $s_0$ from $\pi$.
2. From here onwards, upon entering a new state $i$, sample the time of the next transition from an exponential with rate $|A_{ii}|$, and then a new state $j \neq i$ with probability proportional to $A_{ij}$. These latter two steps are repeated until the end of the interval, giving a piecewise-constant trajectory consisting of a sequence of holds and jumps.

Note that under this formulation, it is impossible for the system to make self-transition, these are effectively absorbed into the rate parameters $A_{ii}$.

---

[*]Equal contribution
[†]Now at Facebook

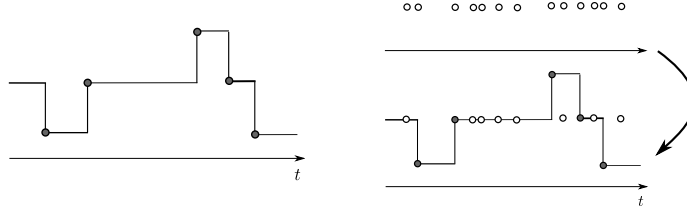

Figure 1: (left) a realization of an MJP, (right) sampling a path via uniformization.

**Bayesian inference for MJPs:** In practical applications, one only observes the MJP trajectory $S(t)$ indirectly through a noisy observation process. Abstractly, this forms a hidden Markov model problem, now in continuous time. For instance, the states of the MJP could correspond to different states of a gene-network, and rather than observing these directly, one only has noisy gene-expression level measurements. Alternately, each state $i$ can have an associated *emission rate* $\lambda_i$, and rather than directly observing $S(t)$ or $\lambda_{S(t)}$, one observes a realization of a Poisson process with intensity $\lambda_{S(t)}$. The Poisson events could correspond to mutation events on a strand of DNA, with position indexed by $t$ (Fearnhead and Sherlock, 2006). In this work, we consider a dataset of users logging their activity into the social media website FourSquare, with each 'check-in' consisting of a time and a location. We model each user with an MJP, with different states having different distributions over check-in locations. Given a sequence of user check-ins, one is interested in quantities like the latent state of the user, various clusters of check-in locations, and the rate at which users transition from one state to another. We describe this problem and the dataset in more detail in our experiments.

In typical situations, the parameters $\pi$ and $A$ are themselves unknown, and it is necessary to learn these, along with the latent MJP trajectory, from the observed data. A Bayesian approach places a prior over these parameters and uses the observed data to obtain a posterior distribution. A simple and convenient prior over $A$ is a Dirichlet-Gamma prior: this places a Dirichlet prior over $\pi$, and models the off-diagonal elements $A_{ij}$ as draws from a Gamma$(a, b)$ distribution. The negative diagonal element $|A_{ii}|$ is then just the sum of the corresponding elements from the same row, and is marginally distributed as a Gamma$((N-1)a, b)$ variable. This prior is convenient in the context of MCMC sampling, allowing a Gibbs sampler that alternately samples $(\pi, A)$ given a MJP trajectory $S(t)$, and then a new trajectory $S(t)$ given $A$ and the observations. The first step is straightforward: given an MJP trajectory, the Dirichlet-Gamma prior is conjugate, resulting in a simple Dirichlet-Gamma posterior (but see Fearnhead and Sherlock (2006) and the next section for a slight generalization that continues to be conditionally conjugate). Similarly, recent developments in MCMC inference have made the second step fairly standard and efficient, see Rao and Teh (2014); Hajiaghayi et al. (2014). Despite its computational simplicity, this Gibbs sampler comes at a price: it can mix slowly due to coupling between $S(t)$ and $A$. Alternate approaches like particle MCMC (Andrieu et al., 2010) do not exploit the MJP stucture, resulting in low acceptance rates, and estimates with high variance. These challenges associated with MCMC raise the need for new techniques for Bayesian inference. Here, we bring recent ideas from variational Bayes towards posterior inference for MJPs, proposing a novel and efficient collapsed variational algorithm that marginalizes out the parameter $A$, thereby addressing the issue of slow mixing. Our algorithm adaptively finds regions of low and high transition activity, rather than integrating these out. In our experiments, we show that these can bring significant computational benefits. Our algorithm is based on an alternate approach to sampling an MJP trajectory called *uniformization* (Jensen, 1953), which we describe next.

## 2 Uniformization

Given a rate matrix $A$, choose an $\Omega > \max |A_{ii}|$, and sample a set of times from a Poisson process with intensity $\Omega$. These form a random discretization of time, giving a set of candidate transition times (Figure 1, top right). Next sample a piecewise-constant trajectory by running a discrete-time Markov chain over these times, with Markov transition matrix given by $B = (I + \frac{1}{\Omega}A)$, and with initial distribution $\pi$. It is easy to verify that $B$ is a valid transition matrix with at least one non-zero diagonal element. This allows the discrete-time system to move back to the same state, something impossible under the original MJP. In fact as $\Omega$ increases the probability of self-transitions increases; however at the same time, a large $\Omega$ implies a large number of Poisson-distributed candidate times. Thus the self-transitions serve to discard excess candidate times, and one can show (Jensen, 1953; Rao and Teh, 2014) that after discarding the self-transitions, the resulting distribution over trajectories is identical to an MJP with rate matrix $A$ for any $\Omega \geq \max |A_{ii}|$ (Figure 1, bottom right).

Rao and Teh (2012) describe a generalization, where instead of a single $\Omega$, each state $i$ has its own dominating rate $\Omega_i > |A_{ii}|$. The transition matrix $B$ is now defined as $B_{ii} = 1 + A_{ii}/\Omega_i$, and

$B_{ij} = A_{ij}/\Omega_i$, for all $i, j \in (1, \ldots, N), i \neq j$. Now, on entering state $i$, one proposes the the next candidate transition time from a rate-$\Omega_i$ exponential, and then samples the next state from $B_i$. As before, self-transitions amount to rejecting the opportunity to leave state $i$. Large $\Omega_i$ result in more candidate transition times, but more self-transitions. Rao and Teh (2012) show that these two effects cancel out, and the resulting path, after discarding self-transitions is a sample from an MJP.

**An alternate prior on the parameters of an MJP:** We use uniformization to formulate a novel prior distribution over the parameters of an MJP; this will facilitate our later variational Bayes algorithm. Consider $A_i$, the $i$th row of the rate matrix $A$. This is specified by the diagonal element $A_{ii}$, and the vector $B_i := \frac{1}{|A_{ii}|}(A_{i1}, \cdots, A_{i,i-1}, 0, A_{i,i+1}, \cdots, A_{iN})$. Recall that the latter is a probability vector, giving the probability of the next state after $i$. In Fearnhead and Sherlock (2006), the authors place a Gamma prior on $|A_{ii}|$, and what is effectively, a Dirichlet$(\alpha, \cdots, 0, \cdots, \alpha)$ prior on $B_i$ (although they treat $B_i$ as an $N-1$-component vector by ignoring the 0 at position $i$).

We place a Dirichlet$(a, \cdots, a_0, \cdots, a)$ prior on $B_i$ for all $i$. Such $B_i$'s allow self-transitions, and form the rows of the transition matrix $B$ from uniformization. Note that under uniformization, the row $A_i$ is uniquely specified by the pair $(\Omega, B_i)$ via the relationship $A_i = \Omega(B_i - 1_i)$, where $1_i$ is the indicator for $i$. We complete our specification by placing a Gamma prior over $\Omega$.

Note that since the rows of $A$ sum to 0, and the rows of $B$ sum to 1, both matrices are completely determined by $N(N-1)$ elements. On the other hand, our specification has $N(N-1) + 1$ random variables, the additional term arising because of the prior over $\Omega$. Given $A$, $\Omega$ plays no role in the generative process defined by Gillespie's algorithm, although it is an important parameter in MCMC inference algorithms based on uniformization. In our situation, $B$ represents transition probabilities *conditioned on there being a transition*, and now $\Omega$ does carry information about $A$, namely the distribution over event times. Later, we will look at the implied marginal distribution over $A$. First however, we consider the generalized uniformization scheme of Rao and Teh (2012). Here we have $N$ additional parameters, $\Omega_1$ to $\Omega_N$. Again, under our model, we place Gamma priors over these $\Omega_i$'s, and Dirichlet priors on the rows of the transition matrix $B$.

Note that in Rao and Teh (2014, 2012), $\Omega$ is set to $2 \max_i |A_{ii}|$. From the identity $B = I + \frac{1}{\Omega}A$, it follows that under any prior over $A$, with probability 1, the smallest diagonal element of $B$ is $1/2$. Our specification avoids such a constrained prior over $B$, instead introducing an additional random variable $\Omega$. Indeed, our approach is equivalent to a prior over $(\Omega, A)$, with $\Omega = k \max_i A_{ii}$ for some *random $k$*. We emphasize that the choice of this prior over $k$ does not effect the generative model, only the induced inference algorithms such as Rao and Teh (2014) or our proposed algorithm.

To better understand the implied marginal distribution over $A$, consider the representation of Rao and Teh (2012), with independent Gamma priors over the $\Omega_i$'s. We have the following result:

**Proposition 1.** *Place independent Dirichlet priors on the rows of $B$ as above, and independent Gamma$((N-1)a + a_0, b)$ priors on the $\Omega_i$. Then, the associated matrix $A$ has off-diagonal elements that are marginally Gamma$(a, b)$-distributed, and negative-diagonal elements that are marginally Gamma$((N-1)a, b)$-distributed, with the rows of $A$ adding to 0 almost surely.*

The proposition is a simple consequence of the Gamma-Dirichlet calculus: first observe that the collection of variables $\Omega_i B_{ij}$ is a vector of independent Gamma$(a, b)$ variables. Noting that $A_{ij} = \Omega_i B_{ij}$, we have that the off-diagonal elements of $A$ are independent Gamma$(a, b)$s, for $i \neq j$. Our proof is complete when we notice that the rows of $A$ sum to 0, and that the sum of independent Gamma variables is still Gamma-distributed, with scale parameter equal to the sum of the scales. It is also easy to see that given $A$, the $\Omega_i$ is set by $\Omega_i = |A_{ii}| + \omega_i$, where $\omega_i \sim$ Gamma$(a_0, b)$.

In this work, we will simply matters by scaling all rows by a single, shared $\Omega$. This will result in a vector of $A_{ij}$'s each marginally distributed as a Gamma variable, but now positively correlated due to the common $\Omega$. We will see that this simplification does not affect the accuracy of our method. In fact, as our variational algorithm will maintain just a point estimate for $\Omega$, so that its effect on the correlation between the $A_{ii}$'s is negligible.

## 3 Variational inference for MJPs

Given noisy observations $X$ of an MJP, we are interested in the posterior $p(S(t), A|X)$. Following the earlier section, we choose an augmented representation, where we replace $A$ with the pair $(B, \Omega)$. Similarly, we represent the MJP trajectory $S(t)$ with the pair $(T, U)$, where $T$ is the set of candidate transition times, and $U$ (with $|U| = |T|$), is the set of states at these times. For our variational

algorithm, we will integrate out the Markov transition matrix $B$, working instead with the marginal distribution $p(T, U, \Omega)$. Such a collapsed representation avoids issues that plague MCMC and VB approaches, where coupling between trajectory and transition matrix slows down mixing/convergence. The distribution $p(T, U, \Omega)$ is still intractable however, and as is typical in variational algorithms, we will make a factorial approximation $p(T, U, \Omega) \approx q(T, U)q(\Omega)$. Writing $q(T, U) = q(U|T)q(T)$, we shall also restrict $q(T)$ to a delta-function: $q(T) = \delta_{\hat{T}}(T)$ for some $\hat{T}$. In this way, finding the 'best' approximating $q(T)$ within this class amounts to finding a 'best' discretization of time. This approach of optimizing over a time-discretization is in contrast to MCMC schemes that integrate out the time discretization, and has a two advantages:

*Simplified computation*: Searching over time-discretization can be significantly more efficient than integrating it out. This is especially true when a trajectory involves bursts of transitions interspersed with long periods of inactivity, where schemes like Rao and Teh (2014) can be quite inefficient.

*Better interpretability*: A number of applications use MJPs as tools to segment a time interval into inhomogeneous segments. A full distribution over such an object can be hard to deal with.

Following work on variational inference for discrete-time Markov chains (Wang and Blunsom, 2013), we will approximate $q(U|T)$ factorially as $q(U|T) = \prod_{t=1}^{|T|} q(u_t)$. Finally, since we fix $q(T)$ to a delta function, we will also restrict $q(\Omega)$ to a delta function, only representing uncertainty in the MJP parameters via the marginalized transition matrix $B$.

We emphasize that even though we optimize over time discretizations, we still maintain posterior uncertainty of the MJP state. Thus our variational approximation represents a distribution over piecewise-constant trajectories as a single discretization of time, with a probability vector over states for each time segment (Figure 2). Such an approximation does not involve too much loss of information, while being more convenient than a full distribution over trajectories, or a set of sample paths. While optimizing over trajectories, our algorithm attempts to find segments where the distribution over states is reasonably constant, if not it will refine a segment into two smaller ones. Our overall variational inference algorithm then involves minimizing the Kullback-Liebler distance between this posterior approximation and the true posterior. We do this in a coordinate-wise manner:

**1) Updating** $q(U|T) = \prod_{t=1}^{|T|} q(u_t)$**:** Given a discretization $T$, and an $\Omega$, uniformization tells us that inference over $U$ is just inference for a discrete-time hidden Markov model. We adapt the approach of Wang and Blunsom (2013) to update $q(U)$. Assume the observations $X$ follow an exponential family likelihood with parameter $C_s$ for state $s$: $p(x_t^l|S_t = s) = \exp(\phi(x_t^l)^T C_s)h(x_t^l)/Z(C_s)$, where $Z$ is the normalization constant, and $x_t^l$ is the l-th observation observed in between $[T_t, T_{t+1}]$. Then for a sequence of $|T|$ observations, we have $p(X, U|B, C) \propto$

$$\prod_{t=0}^{|T|} B_{u_t, u_{t+1}} \prod_{l=1}^{n_t} \exp(\phi(x_t^l)^T C_{u_t})h(x_t^l)/Z(c_{u_t}) = \left[\prod_{i=1}^{S}\prod_{j=1}^{S} B_{ij}^{\#_{ij}}\right] \prod_{i=1}^{S} \exp(\bar{\phi}_i^T C_i)(\prod_{t=0}^{|T|}\prod_{l=1}^{n_t} \frac{h(x_t^l)}{Z(C_{u_t})})$$

Here $n_t$ is the number of observations in $[T_t, T_{t+1}]$ and $\#_{ij}$ is the number of transitions from state $i$ to $j$, and $\tilde{\phi}_t = \sum_{l=1}^{n_t} \phi(x_t^l)$ and $\bar{\phi}_i = \sum_{t, s.t.\ u_t = i} \tilde{\phi}_t$.

Placing Dirichlet($\alpha$) priors on the rows of $B$, and an appropriate conjugate prior on $C$, we have

$$p(X, U, B, C) \propto= \left[\prod_{i=1}^{S}\Gamma(S\alpha)\prod_{j=1}^{S}\frac{B_{ij}^{\#_{ij}+\alpha-1}}{\Gamma(\alpha)}\right]\prod_{i=1}^{S}\exp(C_i^T(\bar{\phi}_i+\beta))(\prod_{t=0}^{|T|}\prod_{l=1}^{n_t}\frac{h(x_t^l)}{Z(C_{u_t})}).$$

Integrating out $B$ and $C$, and writing $\#_i$ for the number of visits to state $i$, we have:

$$p(X, U) \propto= \left[\prod_{i=1}^{S}\frac{\Gamma(S\alpha)}{\Gamma(\#_i+\alpha)}\prod_{j=1}^{S}\frac{\Gamma(\#_{ij}+\alpha)}{\Gamma(\alpha)}\right]\prod_{i=1}^{S}\bar{Z}_i(\bar{\phi}_i+\beta).$$

$$\text{Then, } p(u_t = k|\cdot) \propto \frac{(\#_{u_{t-1},k}^{\neg t}+\alpha)^{\delta_k^t}(\#_{k,u_{t-1}}^{\neg t}+\delta_k^{t-1,t+1}+\alpha)^{\delta_k^t}}{(\#_k^{\neg t}+\alpha)^{\delta_k^t}} \cdot \bar{Z}_k(\bar{\phi}_k^{\neg t}+\bar{\phi}_k(X_t)+\beta)$$

Standard calculations for variational inference give the solution to $q(u_t) = $ argmin KL$(q(U, T, \Omega)\|p(U, T, \Omega|X))$ as $q(u_t) = E_{q^{\neg t}}[\log p(u_t|\cdot)]$, We then have the update

Figure 2: (left) Merging to time segments. (right) splitting a time segment. Horizontal arrows are VB messages.

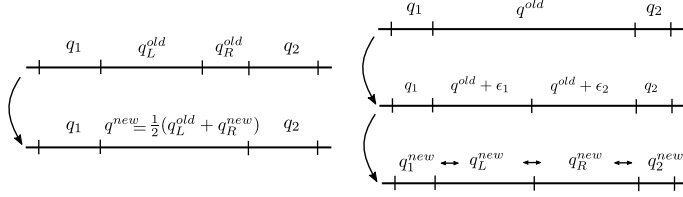

rule:

$$q(u_t = k) \propto \frac{\mathbb{E}_{q^{\neg t}}[\#^{\neg t}_{u_{t-1},k} + \alpha]\mathbb{E}_{q^{\neg t}}[\#^{\neg t}_{k,u_{t-1}} + \delta^{t-1,t+1}_k + \alpha]}{\mathbb{E}_{q^{\neg t}}[\#^{\neg t}_{u_{t-1},k} + S\alpha]\mathbb{E}_{q^{\neg t}}[\#^{\neg t}_{k,u_{t-1}} + \delta^{t-1,t+1}_k + \alpha]} \cdot \frac{\mathbb{E}_{q^{\neg t}}\bar{Z}_k(\bar{\phi}^{\neg t}_k + \bar{\phi}_k(X_t) + \beta)}{\mathbb{E}_{q^{\neg t}}\bar{Z}_k(\bar{\phi}^{\neg t}_k + \beta)}$$

For the special case of multinomial observations, we refer to Wang and Blunsom (2013).

**2) Updating** $q(T)$**:** We perform a greedy search over the space of time-discretizations by making local stochastic updates to the current $T$. Every iteration, we first scan the current $T$ to find a beneficial *merge* (Figure 2, left): go through the transition times in sequential or random order, merge with the next time interval, compute the variational lower bound under this discretization, and accept if it results in an improvement. This eliminates unnecessary transitions times, reducing fragmentation of the segmentation, and the complexity of the learnt model. Calculating the variational bound for the new time requires merging the probability vectors associated with the two time segments into a new one. One approach is to initialize this vector to some arbitrary quantity, run step 1 till the $q$'s converge, and use the updated variational bound to accept or reject this proposal. Rather than taking this time-consuming approach, we found it adequate to set the new $q$ to a convex combination to the old $q$'s, each weighted by the length of their corresponding interval length. In our experiments, we found that this performed comparably at a much lower computational cost.

If no merge is found, we then try to find a beneficial split. Again, go through the time segments in some order, now splitting each interval into two. After each split, compare the likelihood before and after the split, and accept (and return) if the improvement exceeds a threshold. Again, such a split requires computing probability vectors for the newly created segments. Now, we assign each segment the same vector as the original segment (plus some noise to break symmetry). We then run one pass of step 1, updating the $q$'s on either side of the new segment, and then updating the $q$'s in the two segments. We consider two interval splitting schemes, bisection and random-splitting.

Overall, our approach is related to split-merge approaches for variational inference in nonparametric Bayesian models Hughes et al. (2015); these too maintain and optimize point estimates of complex, combinatorial objects, instead maintaining uncertainty over quantities like cluster assignment. In our real-world check-in applications, we consider a situation where there is not just one MJP trajectory, but a number of trajectories corresponding to different users. In this situation, we take a stochastic variational Bayes approach, picking a random user and following the steps outlined earlier.

**Updating** $q(\Omega)$**:** With a Gamma$(a_1, a_2)$ prior over $\Omega$, the posterior over $\Omega$ is also Gamma, and we could set $\Omega$ to the MAP. We found this greedy approach unstable sometimes, instead using a partial update, with the new $\Omega$ equal to the mean of the old value and the MAP value. Writing $s$ for the total number of transition times in all $m$ trajectories, this gives us $\Omega_{new} = (\Omega_{old} + (a_1 + s)/(a_2 + m))/2$.

## 4 Experiments

We present qualitative and quantitative experiments using synthetic and real datasets to demonstrate the accuracy and efficiency of our variational Bayes (VB) algorithm. We mostly focus on comparisons with the MCMC algorithms from Rao and Teh (2014) and Rao and Teh (2012).

**Datasets.** We use a dataset of check-in sequences from 8967 FourSquare users in the year 2011, originally collected by Gao et al. (2012) for studying location-based social networks. Each check-in has a time stamp and a location (latitude and longitude), with users having 191 check-in records on average. We only consider check-ins inside a rectangle containing the United States and parts of Mexico and Canada (see Figure 3, left), and randomly select 200 such sequences for our experiments. We partition the space into a $40 \times 40$ grid, and define the observation distribution of each MJP state as a categorical distribution over the grid cells. See Pan et al. (2016) for more details on this application.

We also use two synthetic datasets in our experiments, with observations in a $5 \times 5$ grid. For the first dataset, we fix $\Omega = 20$ and construct a transition matrix $B$ for 5 states with $B(i, i) = 0.8$,

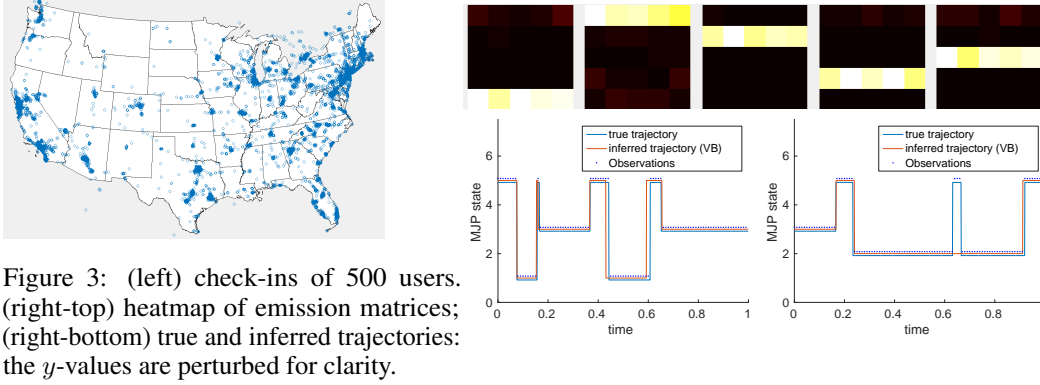

Figure 3: (left) check-ins of 500 users. (right-top) heatmap of emission matrices; (right-bottom) true and inferred trajectories: the $y$-values are perturbed for clarity.

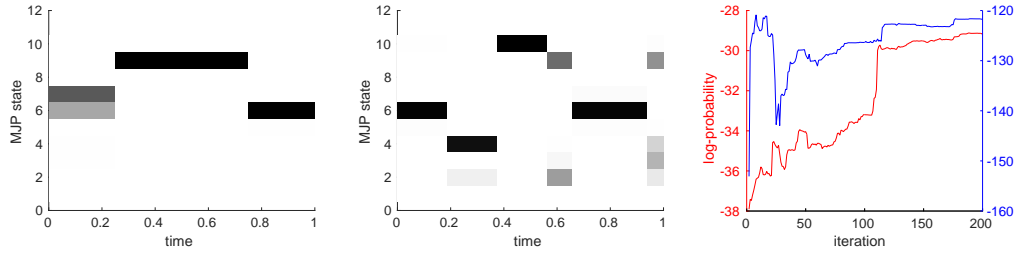

Figure 4: (left,middle) posterior distribution over states of two trajectories in second synthetic dataset; (right) evolution of $\log p(T \mid \Omega, X)$ in the VB algorithm for two sample sequences

$B(i, 5) = 0.19$, $B(5, 5) = 0$, and $B(5, i) = 0.25$ for $i \in [1, 4]$. By construction, these sequences can contain many short time intervals at state 5, and longer time intervals at other states. We set the observation distribution of state $i$ to have 0.2 probability on grid cells in the $i$-th row and 0 probability otherwise. For the second synthetic dataset, we use 10 states and draw both the transition probabilities of $B$ and the observation probabilities from Dirichlet(1) distribution. Given $(\Omega, B)$, we sample 50 sequences, each containing 100 evenly spaced observations.

**Hyperparameters:** For VB on synthetic datasets we place a Gamma(20, 2) prior on $\Omega$, and Dirichlet(2) priors on the transition probabilities and the observation probabilities, while on the check-in data, a Gamma(6, 1), a Dirichlet(0.1) and a Dirichlet(0.01) are placed. For MCMC on synthetic datasets, we place a Gamma(2, 0.2) and a Dirichlet(0.1) for the rate matrix, while on the check-in data, a Gamma(1, 1) and a Dirichlet(0.1) are placed.

**Visualization:** We run VB on the first synthetic dataset for 200 iterations, after which we use the posterior expected counts of observations in each state to infer the output emission probabilities (see Figure 3(top-right)). We then relabel the states under the posterior to best match the true state (our likelihood is invariant to state labels); Figure 3(bottom-right) shows the true and MAP MJP trajectories of two sample sequences in the synthetic dataset. Our VB algorithm recovers the trajectories well, although it is possible to miss some short "bumps". MCMC also performs well in this case, although as we will show, it is significantly more expensive.

The inferred posteriors of trajectories have more uncertainty for the second synthetic dataset. Figure 4 (left and middle) visualizes the posterior distributions of two hidden trajectories with darker regions for higher probabilities. The ability to maintain posterior uncertainty about the trajectory information

Figure 5: reconstruction error of MCMC and VB (using random and even splitting) for the (left) first and (right) the second synthetic dataset. The random split scheme is in blue , even split scheme is in red , and VB random split scheme with true omega in orange. MCMC is in black.

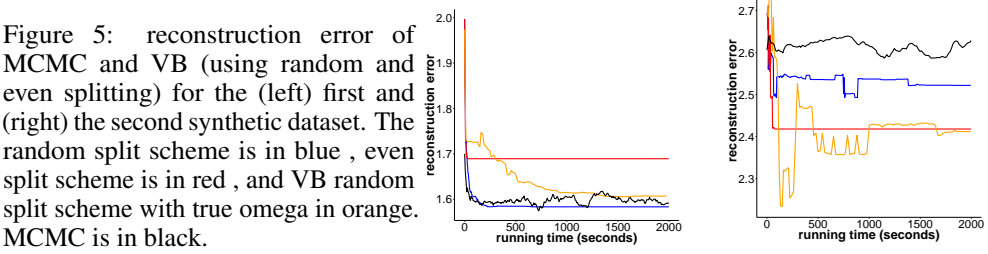

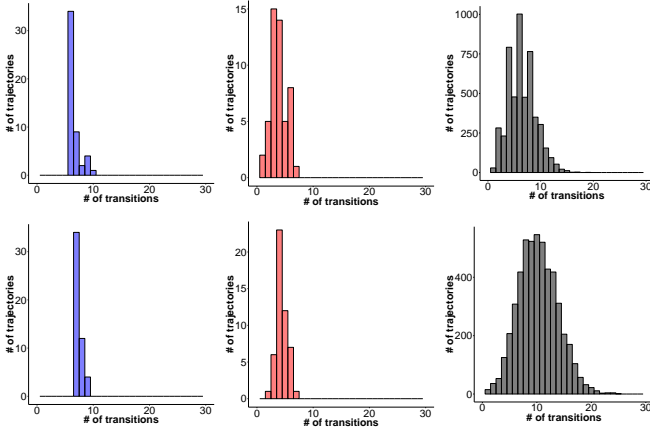

Figure 6: Synthetic dataset 1(top) and 2(bottom): Histogram of number of transitions using VB with (left) random splitting; (middle) even spliting; (right) using MCMC.

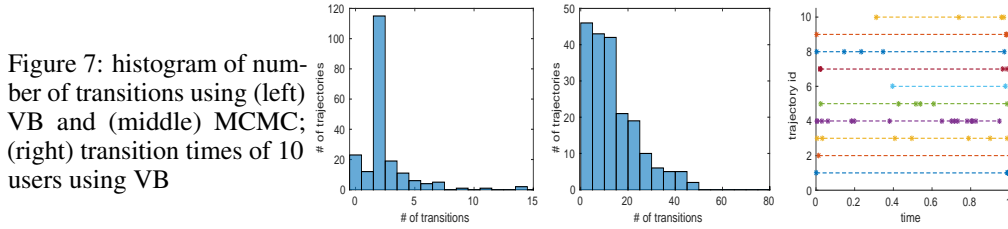

Figure 7: histogram of number of transitions using (left) VB and (middle) MCMC; (right) transition times of 10 users using VB

is important in real world applications, and is something that k-means-style approximate inference algorithms (Huggins et al., 2015) ignore.

**Inferred trajectories for real-world data.** We run the VB algorithm on the check-in data using 50 states for 200 iterations. Modeling such data with MJPs will recover MJP states corresponding to cities or areas of dense population/high check-in activity. We investigate several aspects about the MJP trajectories inferred by the algorithm. Figure 4(right) shows the evolution of $\log p(T \mid \Omega, X)$ (up to constant factor) of two sample trajectories. This value is used to determine whether a merge or split is beneficial in our VB algorithm. It has an increasing trend for most sequences in the dataset, but can sometimes decrease as the trajectory discretization evolves. This is expected, since our stochastic algorithm maintains a pseudo-bound. Figure 6 shows similar results for the synthetic datasets.

Normally, we expect a user to switch areas of check-in activity only a few times in a year. Indeed, Figure 7 (left) shows the histogram of the number of transition times across all trajectories, and the majority of trajectories have 3 or less transitions. We also plot the actual transition times of 10 random trajectories (right). In contrast, MCMC tends to produce more transitions, many of which are redundant. This is a side effect of uniformization in MCMC sampling, which requires a homogeneously dense Poisson distributed trajectory discretization at every iteration.

**Running time vs. reconstruction error.** We measure the quality of the inferred posterior distributions of trajectories using a reconstruction task on the check-in data. We randomly select 100 test sequences, and randomly hold out half of the observations in each test sequence. The training data consists of the observations that are not held out, i.e., 100 full sequences and 100 half sequences. We run our VB algorithm on this training data for 200 iterations. After each iteration, we reconstruct the held-out observations as follows: given a held-out observation at time $t$ on test sequence $\tau$, using the maximum-likelihood grid cell to represent each state, we compute the expected grid distance between the true and predicted observations using the estimated posterior $q(u_t)$. The reconstruction error for $\tau$ is computed by averaging the grid distances over all held-out observations in $\tau$. The overall reconstruction error is the average reconstruction error over all test sequences. Similarly, we run the MCMC algorithm on the training data for 1000 iterations, and compute the overall reconstruction error after every 10 iterations, using the last 300 iterations to approximate the posterior distribution of the MJP trajectories. We also run an improved variant of the MCMC algorithm, where we use the generalized uniformization scheme Rao and Teh (2012) with different $\Omega_i$ for each state. This allows coarser discretizations for some states and typically runs faster per iteration.

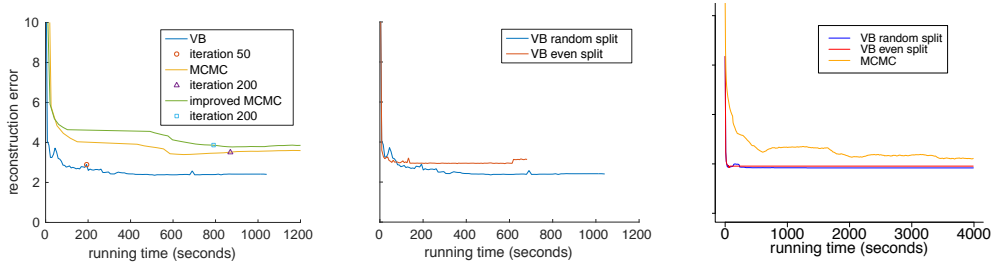

Figure 8: (left) reconstruction error of VB and MCMC algorithms; (middle) reconstruction error using random and even splitting; (right) reconstruction error for more iterations

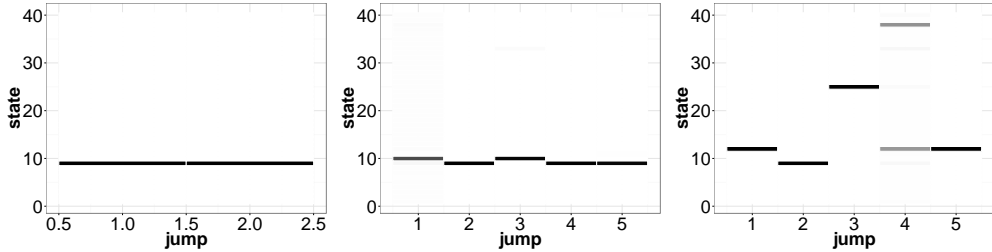

Figure 9: Posterior distribution over states of three trajectories in checkin dataset.

Figure 8(left) shows the evolution of reconstruction error during the algorithms. The error using VB plateaus much more quickly than the MCMC algorithms. The error gap between MCMC and VB is because of slow mixing of the paths and parameters, as a result of the coupling between latent states and observations as well as modeling approximations. Although the improved MCMC takes less time per iteration, it is not more effective for reconstruction in this experiment. Figure 5 shows similar results for the synthetic datasets. Figure 9 visualizes the posterior distributions of three hidden trajectories with darker shades for higher probabilities.

We have chosen to split each time interval randomly in our VB algorithm. Another possibility is to simply split it evenly. Figure 8(middle) compares the reconstruction error of the two splitting schemes. Random splitting has lower error since it produces more successful splits; on the other hand, the running time is smaller with even splitting due to fewer transitions in the inferred trajectories. In Figure 8(right), we resampled the training set and the testing set and ran the experiment for longer. It shows that the error gap between VB and MCMC is closing.

**Related and future work**: Posterior inference for MJPs has primarily been carried out via MCMC Hobolth and Stone (2009); Fearnhead and Sherlock (2006); Bladt and Sørensen (2005); Metzner et al. (2007). The state-of-the-art MCMC approach is the scheme of Rao and Teh (2014, 2012), both based on uniformization. Other MCMC approaches center around particle MCMC Andrieu et al. (2010), e.g. Hajiaghayi et al. (2014). There have also been a few deterministic approaches to posterior inference. The earliest variational approach is from Opper and Sanguinetti (2007), although they consider a different problem from ours, viz. structured MJPs with interacting MJPs (e.g. population sizes of a predator and prey species, or gene networks). They then use a mean-field posterior approximation where these processes are assumed independent. Our algorithm focuses on a single, simple MJP, and an interesting extension is to put the two schemes together for systems of coupled MJPs. Finally a recent paper Huggins et al. (2015) that studies the MJP posterior using a small-variance asymptotic limit. This approach, which generalizes k-means type algorithms to MJPs however provides only point estimates of the MJP trajectory and parameters, and cannot represent posterior uncertainty. Additionally, it still involves coupling between the MJP parameters and trajectory, an issue we bypass with our collapsed algorithm.

There are a number of interesting extensions worth studying. First is to consider more structured variational approximations (Wang and Blunsom, 2013), than the factorial approximations we considered here. Also of interest are extensions to more complex MJPs, with infinite state-spaces (Saeedi and Bouchard-Côté, 2011) or structured state-spaces (Opper and Sanguinetti, 2007). It is also interesting to look at different extensions of the schemes we proposed in this paper: different choices of split-merge proposals, and more complicated posterior approximations of the parameter $\Omega$. Finally, it is instructive to use other real-world datasets to compare our approaches with more traditional MCMC approaches.

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
