[Reviews · NeurIPS 2017]

Reviewer 1



This paper proposes performing Bayesian inference for Markov jump processes using a collapsed variational inference algorithm. Specifically, the authors collapses out the transition times and instead optimizes a discretization of time allowing for easy updates of the remaining model parameters. The method is applied to analyzing user check-in data on FourSquare. The main ideas of the paper and the variational inference algorithm are very interesting and the authors do a good job explaining them. There are two aspects of the paper that I have questions about. First, the paper does not actually show that variational algorithm is necessary. Specifically, the experiments do not clearly indicate where uncertainty quantification from the variational approximation is useful. In other words, why would someone use this algorithm over say just performing MAP inference? Second, the authors propose a greedy method to optimize the time discretization that works by splitting and merging segments of time and using the ELBO to accept the update. This would require that the ELBOs are nested so that their values are comparable. It is not clear to me that this is the case and the authors should definitely include a justification for this. Finally, as stated before, the experiments are focused on reconstructing paths and it isn't clear why having a posterior distribution is useful. Beyond these major concerns I have some minor comments: - There are a lot of typos (some of which are noted here) - Line 82: "bottom right" -> "right" - In the "Updating q(U|T)" section: I think you mean "exponential family likelihood", not "exponential likelihood". - \bar{Z}_i was never defined Overall, I think that this paper could be accepted. I would just like the authors to address my major concerns above.

Reviewer 2



The authors present a variational inference algorithm for continuous time Markov jump processes. Following previous work, they use "uniformization" to produce a discrete time skeleton at which they infer the latent states. Unlike previous work, however, the authors propose to learn this skeleton (a point estimate, via random search) and to integrate out, or collapse, the transition matrix during latent state inference. They compare their algorithm to existing MCMC schemes, which also use uniformization, but which do not collapse out the transition matrix. While this work is well motivated, I found it difficult to tease out which elements of the inference algorithm led to the observed improvement. Specifically, there are at least four dimensions along which the proposed method differs from previous work (e.g. Rao and Teh, 2014): (i) per-state \Omega_i vs shared \Omega for all states; (ii) learned point estimate of discretization vs sampling of discretization; (iii) variational approximation to posterior over latent states vs sampling of latent state sequence; and (iv) collapsing out transition matrix vs maintaining sample / variational factor for it. That the "Improved MCMC" method does not perform better than "MCMC" suggests that (i) does not explain the performance gap. It seems difficult to test (ii) explicitly, but one could imagine a similar approach of optimizing the discretization using sample based estimates of the marginal likelihood, though this would clearly be expensive. Dimensions (iii) and (iv) suggest two natural alternatives worth exploring. First, an "uncollapsed" variational inference algorithm with a factor for the transition matrix and a structured variational factor on the complete set of latent states U. Indeed, given the discretization, the problem essentially reduces to inference in an HMM, and uncollapsed, structured variational approximation have fared well here (e.g. Paisley and Carin, 2009; Johnson and Willsky, 2014). Second, it seems you could also collapse out the transition matrix in the MCMC scheme of Rao and Teh, 2014, though it would require coordinate-wise Gibbs updates of the latent states u_t | u_{\neg t}, just as your proposed scheme requires q(U) to factorize over time. These two alternatives would fill in the gaps in the space of inference algorithms and shed some light on what is leading to the observed performance improvements. In general, collapsing introduces additional dependencies in the model and precludes block updates of the latent states, and I am curious as to whether the gains of collapsing truly outweigh these costs. Without the comparisons suggested above, the paper cannot clearly answer this question. Minor comments: - I found the presentation of MCMC a bit unclear. In some places (e.g. line 252) you say MCMC "requires a homogeneously dense Poisson distributed trajectory discretization at every iteration." but then you introduce the "Improved MCMC" algorithm which has an \Omega_i for each state, and presumably has a coarser discretization. References: Paisley, John, and Lawrence Carin. "Hidden Markov models with stick-breaking priors." IEEE Transactions on Signal Processing 57.10 (2009): 3905-3917. Johnson, Matthew, and Alan Willsky. "Stochastic variational inference for Bayesian time series models." International Conference on Machine Learning. 2014.

Reviewer 3



The paper proposes an algorithm for variational Bayesian inference in continuous-time Markov jump processes. In particular, while other inference methods for these models make separate updates to the inferred transition matrix and the inferred trajectory and hence suffer from the strong coupling between the two, the proposed method aims to marginalize out the transition matrix and only update the trajectory (and a uniformizing variable). The basic idea here seems like a good one, though there are some significant weaknesses to the current version of the paper. The updates to the inferred transition times (the variational factor q(T), which is a point estimate) are greedy and heuristic. The paper would be made stronger if these heuristic updates were studied more closely and in isolation. Can the procedure get stuck in poor local optima? Are there conditions under which it should work well, or tricks to its initialization? It would be informative to have more detailed experiments on this step alone, though Figure 6 is a start. Using an unstructured variational factor on q(U|T) is also a weakness, and the update is very similar to Algorithm 1 of Wang and Blunsom (2013). As a result, I think the main algorithmic contribution is in the update to q(T), which could probably be improved. The plots look a bit rushed. Some text labels are hard to read and some legends are missing. Also, in several plots the MCMC method does not achieve the same error levels as the VB methods; this error gap is explained in terms of slow mixing (lines 268-271), but in that case it would be informative to run the MCMC methods for longer and see the gap close, or explain why that's not possible. In addition, some of the example trajectories look like toy problems. Given the heuristic nature of the main algorithmic contribution and the weaknesses in the plots and experiments presented, I think this is a borderline paper.